# Further Delineation of Clinical Phenotype of *ZMYND11* Variants in Patients with Neurodevelopmental Dysmorphic Syndrome

**DOI:** 10.3390/genes15020256

**Published:** 2024-02-19

**Authors:** Aleksandra Bodetko, Joanna Chrzanowska, Malgorzata Rydzanicz, Agnieszka Borys-Iwanicka, Pawel Karpinski, Joanna Bladowska, Rafal Ploski, Robert Smigiel

**Affiliations:** 1Department of Pediatrics, Endocrinology, Diabetology and Metabolic Diseases, Wroclaw Medical University, 50-368 Wroclaw, Poland; ola.bodetko@gmail.com (A.B.); robert.smigiel@umw.edu.pl (R.S.); 2Department of Medical Genetics, Medical University of Warsaw, 02-106 Warsaw, Poland; malgorzata.rydzanicz@wum.edu.pl (M.R.); rploski@wp.pl (R.P.); 3Department of Paediatrics, Gastroenterology and Nutrition, Wroclaw Medical University, 50-369 Wroclaw, Poland; 4Department of Genetics, Wroclaw Medical University, 50-368 Wroclaw, Poland; pawel.karpinski@umw.edu.pl; 5Department of Radiology, Wroclaw 4th Military Clinical Hospital, Faculty of Medicine, Wroclaw University of Science and Technology, 53-114 Wroclaw, Poland; joanna.bladowska@pwr.edu.pl; 6Department of Radiology and Imaging Diagnostics, Emergency Medicine Center, Marciniak Lower Silesian Specialist Hospital, 54-049 Wroclaw, Poland

**Keywords:** *ZMYND11*, neurodevelopmental dysmorphic syndrome, hyperinsulinaemic hypoglycaemia, diazoxide

## Abstract

Intellectual disability with speech delay and behavioural abnormalities, as well as hypotonia, seizures, feeding difficulties and craniofacial dysmorphism, are the main symptoms associated with pathogenic variants of the *ZMYND11* gene. The range of clinical manifestations of the ZMYND phenotype is constantly being expanded by new cases described in the literature. Here, we present two previously unreported paediatric patients with neurodevelopmental challenges, who were diagnosed with missense variants in the *ZMYND11* gene. It should be noted that one of the individuals manifested with hyperinsulinaemic hypoglycaemia (HH), a symptom that was not described before in published works. The reason for the occurrence of HH in our proband is not clear, so we try to explain the origin of this symptom in the context of the *ZMYND11* syndrome. Thus, this paper contributes to knowledge on the range of possible manifestations of the *ZMYND* disease and provides further evidence supporting its association with neurodevelopmental challenges.

## 1. Introduction

Abnormalities in the processes of epigenetic regulation undoubtedly disrupt physiological growth progression of the human brain and contribute to the aetiology of neurodevelopmental disorders. *ZMYND11* encodes a zinc finger MYND domain-containing protein that acts as a transcriptional co-repressor to inhibit the elongation phase of RNA polymerase II via the recognition of histone modification of the transcribed regions. The *ZMYND11* gene has been reported to play a crucial role in the 10p15.3 microdeletion syndrome, which is associated with neurodevelopmental disorders, dysmorphic features, hypotonia and seizures [1,2]. Heterozygous pathogenic variants in the *ZMYND11* gene have been subsequently identified as the causes of autosomal dominant intellectual developmental disorder-30 with speech delay and behavioural abnormalities (MRD30, an intellectual developmental disorder, autosomal dominant 30, OMIM 616083, ORPHA 178469). Additional features may include various types of seizures, hypotonia, feeding difficulties and craniofacial dysmorphism. This phenotype resembles the 10p15.3 microdeletion syndrome [2,3]. However, in 2012, in a group of 343 children with autism spectrum disorder (ASD), a splice variant of the *ZMYND11* gene was identified in patients with ASD but without intellectual disability and obvious dysmorphism [4,5]. In 2020, Yates et al. described 27 patients (including 16 previously unreported individuals) with pathogenic variants in the *ZMYND11* gene, suggesting a genotype–phenotype correlation [6]. In 2021, Oates et al. identified 47 people (including 16 previously unreported patients) with pathogenic variants in the *ZMYND11* gene and described in detail 20 patients with epilepsy (including 11 previously unreported patients). It was found that neurodevelopmental disorders were common in people with epilepsy (intellectual disability: mild to moderate in 16/20 individuals, and severe in 4/20 individuals). Dysmorphic features were variable and occurred in only 12 patients (12/20) [7].

Here, we present two previously unreported paediatric patients with neurodevelopmental dysmorphic syndrome due to pathogenic variants in *ZMYND11*.

## 2. Clinical Report

### 2.1. Patient 1

The first patient is currently a 5-year-and-a-half-old boy born from non-consanguineous, healthy Caucasian parents. His main clinical features include developmental delay (especially affecting speech), dysmorphic features and a short stature.

He was born at the 37th week of gestation by spontaneous vaginal delivery, with a birth weight of 4400 g (SDS 2.39), a length of 57 cm (SDS 4.03), a head circumference of 37 cm (SDS 2.14) and a score of 10 points on the Apgar scale. During the neonatal period, an episode of transient hypoglycaemia (31 mg/dL and 40 mg/dL) occurred and, therefore, intravenous intake of 10% glucose was required for three days. His hearing assessment was normal. In general, respiratory infections occurred repeatedly throughout childhood. Otitis media occurred at 18 months of age. His development appeared delayed. The boy underwent rehabilitation. He was able to sit since he was 24 months old, and able to walk since he was 4 years old. Tooth eruption was slightly delayed.

At the age of 6 months, due to delayed psychomotor development, the boy was hospitalised in the department of paediatric neurology. It was noted that hypotonia occurred since 2 months of life. From the third month of age, an episode of absent staring was noted. An EEG examination showed no deviations. In the 14th month of age, this patient was hospitalised due to fever and an episode of seizures. When in hospital, he was diagnosed with meningitis and encephalitis during the course of a Streptococcus pneumoniae infection, which was treated with vancomycin, cefotaxime, dexamethasone, furosemide and mannitol. Levetiracetam and sodium valproate were recommended to treat seizures. He was treated with sodium valproate for only 3 months. Levetiracetam was discontinued when he was 4 years old. He had no seizures while taking levetiracetam, but a history of anxiety attacks with awakenings at night was noted. Until the age of 5, he had severe sleep problems and woke up restless. The patient stayed under ophthalmological care due to strabismus and nystagmus.

The brain MRI examination performed at the age of 6 months (Figure 1A,B) revealed a significantly enlarged cisterna magna and moderate dilatation of sulci in the frontal lobes, as well as moderate dilatation of the lateral and third ventricles as a feature of cortico-subcortical atrophy with an adjacent enlargement of the subarachnoid space in the fronto-temporal areas. There was also delayed myelination of the white matter, indicating a typical pattern for the age of 4 months (a high signal intensity of the splenium of the corpus callosum on the T1-weighted image). A follow-up MRI examination was performed at the age of 14 months (Figure 1C,D) and showed progression of the myelination process. However, there were still features of cortico-subcortical atrophy. Moreover, the cortex/white matter differentiation was blurred, especially on the T2-weighted image, instead of being clearly visible at this age, which could indicate a demyelination disorder.

From the moment of delivery, sucking and swallowing disorders, regurgitation and anxiety attacks were observed. Up to 4 months of age, the patient was breastfed. During the infancy period, based on the clinical picture (abdominal pain, bloating, regurgitation, anxiety and constipation), allergy to cow proteins was diagnosed and an amino acid-based hypoallergenic formula was applied. Due to constipation, he was treated with macrogol. Since the age of 2, he has remained on a low fermentable oligosaccharide, disaccharide, monosaccharide and polyol (FODMAP) diet (including being gluten-free).

At the age of 3 years and 2 months, the boy was hospitalised in the department of paediatric endocrinology due to hypoglycaemia with transient hyperinsulinaemia (the outpatient laboratory tests showed glycaemia at 54 mg/dL with insulin at 75.1 μIU/mL, and fasting glycaemia at 49 mg/dL [normal range: 60–99 mg/dL] with insulin at 6.8 μIU/mL [normal range: <24.9 μIU/mL]). No hypoglycaemia was observed during hospitalisation. The HbA1c level was 4.6% (normal range: 4.8–5.9%). It was concluded that the cause of lower glycaemia could possibly be a too restrictive diet (a low FODMAP diet).

At the age of 5 years and a half, a physical examination performed by a clinical geneticist revealed facial/cranial dysmorphic features, including an asymmetry of the skull with bitemporal narrowing and prominent frontal tubercles, long and thick eyebrows, deep-set eyes, strabismus, low-set ears with prominent auricles, and a consistently open mouth. Other features included a short nose, a low hairline, hyperelastic skin, a short neck, a prominent pectus carinatum, raised shoulder blades and kyphosis. Moreover, hypotonia, an abnormal wide-based gait and a short stature were noted. The patient gives the general impression of an overactive child with increased mobility. He is nonverbal but makes sounds. His weight at 5 year of age is 17.3 kg, his height is 101.7 cm (<3rd percentile, HSDS-3.43), his BMI is 16.7 kg/m^2^ (75–90th percentile), and the occipito-frontal circumference (OFC) is 53.8 cm (75–90th percentile).

His first genetic consultation was at the age of 6 months because of developmental delay and dysmorphic features. The facial phenotype of patient 1 is presented in Figure 2. His array CGH did not reveal any copy number changes. At the age of 12 months, further genetic diagnostics was performed using NGS-based whole-exome sequencing (WES) in a singleton scheme (only the proband) and with Sanger verification in both the proband and his parents.

### 2.2. Patient 2

The second patient is currently a 30-month-old girl born from non-consanguineous, healthy Caucasian parents. Her main clinical features include severe global developmental delay, feeding problems, dysmorphic features and hyperinsulinaemic hypoglycaemia.

She was born at the 38th week of gestation by spontaneous vaginal delivery, with a birth weight of 4200 g (SDS 1.94), a length of 58 cm (SDS 4.77), a head circumference of 34 cm (SDS 0) and a score of 6/7 points on the Apgar scale. An umbilical cord gas analysis showed a pH of 7.4 and a BE of 0.6. In the physical examination of the newborn, numerous petechiae on the head and trunk were observed. During the neonatal period, there were pneumonia, respiratory failure and thrombocytopenia. An episode of transient hypoglycaemia (12 mg/dL [normal range: 60–99 mg/dL]) occurred 16 h after delivery. On the third day of life, a CT scan of the abdomen and head was performed, and it led to the suspicion of a spleen injury and hypoxic and post-traumatic lesions of the brain (blood/haematoma on the surface of the cerebellar tentorium on the right side, and suspicion of indentation injuries to the skull bones). Moreover, a transfontanellar ultrasound showed calcifications along the course of the lenticulostriatal vessels. At 3 months of age, the blood test for CMV IgM and IgG antibodies was positive, and the genetic material of the virus was also present in the urine (detected by PCR). Nevertheless, the test was performed in the third month of the child’s life, which made it impossible to determine whether the infection was intrauterine or postnatal. At 8 months of age, a hearing test based on wave V yielded a bilateral threshold of 30 dBnHL at 2–4 kHz (mild hearing loss). Recurrent lower respiratory tract infections occurred throughout the first year of life, often presenting with shortness of breath and respiratory failure. For this reason, hospitalisation in the intensive care unit was required twice. A noticeable gross motor and fine motor delay was observed (she was able to roll over, but not yet able to achieve the ability to sit). Speech was absent. There was a significant delay in tooth eruption (first teeth at 12 months of age).

From the age of 2 months onwards, seizure incidents of various morphologies without fever and episodes of unconsciousness were observed. A sleep EEG performed at the age of 2.5 months showed bifocal seizure activity, which led to the application of a phenobarbital treatment. No epileptic seizures were observed from the commencing of the medication administration up till the 13th month of life. Afterwards, several incidents of immobility with eye rotation and increased muscle tone were noted, and an epileptic episode with the morphology of limb tremor with eye rotation to the right and mouth open occurred. A sleep EEG performed at the age of 14 months showed background disorganization and epileptiform discharges. Seizures remained intractable despite treatment with antiepileptic drugs, such as lamotrigine and phenobarbital.

A brain MRI examination performed at the age of 14 months revealed an absence of normal myelination according to the age of the child. The T1- and T2-weighted images (Figure 3A,B) suggested delayed myelination of the cerebral white matter. The MRI appearance indicated a stage of myelination that was typical for the age of 8 months as there was a pronounced difference between the “anterior” and “posterior” parts of the brain, which was visible on the T1-weighted image with a lower signal within the former one. However, cortex/white matter differentiation could be detected instead of being blurred, which is a typical finding of a normal myelination process, meaning that a dysmyelination disorder should be suspected in this child.

There was also dilatation of the lateral ventricles, the third ventricle and the cerebral sulci as a sign of cerebral cortico-subcortical atrophy. The MRI also showed a cavum septum pellucidum, which is a normal variant of the CSF space, between the leaflets of the septum pellucidum (Figure 3C). The SWI sequence (Figure 3D) did not reveal any low signal of hemosiderin depositions, which meant that there were no signs of any previous intracranial bleeding, including intraventricular haemorrhage (IVH).

The patient was initially fed by a nasogastric tube with breast milk, then with a first infant formula. Abdominal pain and constipation suggested food allergy. After modifying the diet (hydrolysed whey protein) and pharmacological treatment (macrogol, trimebutine and omeprazole), a partial improvement was observed. At the age of 11 months, the patient was admitted to the gastroenterological ward due to feeding problems, such as food retraction, spitting up and gag reflex. The patient’s weight was 10.26 kg, with a length of 74 cm (weight-to-length ratio at 90–97th percentile). At 12 months of age, due to a gastroesophageal reflux disease (GERD) as well as recurrent aspiration pneumonia, a Nissen fundoplication was performed and percutaneous endoscopic gastrostomy (PEG) was placed. Until now, the patient is under the care of the nutrition ward. Currently, she is fed through a gastrostomy with a liquid, high-calorie diet (hydrolysed whey protein). Due to orofacial disorders, neurologopedic rehabilitation was recommended. In the second year of life, a reduction in the incidence of lower respiratory tract infections was observed (pneumonia and bronchitis), although she suffered from purulent otitis three times during this period.

Hypoglycaemia of 11 mg/dL was noted during a seizure attack in the 15 months of age. Therefore, she was diagnosed in the department of paediatric endocrinology. Glucose levels were tested successively from the moment of birth and, except for the first day of life, the results were normal. Hypoglycaemia reappeared 2 days before the percutaneous gastrostomy placement at the age of 12 months (glycaemia at 30 mg/dL and 32 mg/dL, control at 121 mg/dL). Thereafter, until 15 months of age, glycaemia was not checked. On admission, a physical examination revealed moderate macroglossia, the presence of two teeth, generalised hypotonia, and a noticeable gross and fine motor delay. The auxological examination found a weight of 11.5 kg, a length of 83.5 cm (>90th percentile, HSDS 1.76), and a weight-to-length ratio at the 75th percentile. Hyperinsulinaemic hypoglycaemia was confirmed (critical sample at the time of hypoglycaemia: glycaemia at 21 mg/dL and insulin at 21.9 uIU/mL [normal value during hypoglycaemia: <2 μIU/mL], C-peptide at 2.2 ng/mL [normal value during hypoglycaemia <0.5 ng/mL], and urine ketones were not detected. Additionally, appropriate counterregulatory hormone response was as follows: growth hormone at 10.7 ng/mL [normal value during hypoglycaemia: >7 ng/mL], cortisol at 20 ug/dl [normal value during hypoglycaemia: >20 μg/dl]), HbA1c at 4.7% (normal range: 4.8–5.9%), IGF1 at 33.2 ng/mL (normal range: 55–327 ng/mL), and IGFBP-3 at 2.21 ug/mL (normal range: 0.7–3.6 ug/mL)). The abdominal ultrasonography showed hepatomegaly. Echocardiography was normal. The patient was fed via gastrostomy regularly every 3 h with extensively hydrolysed whey protein that was suitable as a sole source of nutrition since she was unable to take solid food. The girl was assessed with continuous glucose monitoring. On the first glucose readings, a certain rhythmicity with fluctuations, including hypoglycaemia, was observed (Figure 4A). Diazoxide was initiated (5 mg/kg/day), and the dose was titrated to 10.8 mg/kg/day that was divided into three doses. Afterwards, no seizures or episodes of hypoglycaemia (Figure 4B) were observed, so the patient was responsive to diazoxide. From the 18th month of age, an increase in the frequency of gag reflexes was noticeable. There was a tendency towards a mild hyperkalaemia (before treatment with diazoxide, potassium level ranged from 4.1 to 6.9; after the introduction of diazoxide, potassium level ranged from 4.2 to 6.1 mmol/l; N: 3.8–5.5).

A physical examination at the age of 22 months performed by a clinical geneticist revealed facial/skull dysmorphic features resembling Cornelia de Lange syndrome, such as small teeth, a consistently open mouth, a low hairline, prominent eyebrows, and increased hairiness on the lower legs, thighs and forearms. The facial phenotype of patient 2 is presented in Figure 5. Excessive skin between the fingers of the hands, proximal displacement of the thumb, strabismus, nystagmus, hypotonia and increased excitability were also observed. The patient’s breathing was accompanied by wheezing. Her weight was 13.2 kg, her height was 88 cm (HSDS 1.36), and the OFC was 45.5 cm (<3rd percentile). At the age of 30 months, her current weight is 11.9 kg, her height is 90 cm (25–50th percentile, HSDS-0.53), and her BMI is 14.69 kg/m^2^ (15–25th percentile according to the WHO).

Due to the suggestive features of Beckwith–Wiedemann syndrome (BWS), such as hyperinsulinaemic hypoglycaemia (HH) and macroglossia, molecular testing was performed (MS-MLPA), with a negative result. Further analysis of the phenotype by the clinical geneticist did not suggest continuing the diagnosis towards BWS. At the age of 20 months, further genetic diagnostics was performed using NGS-based whole-exome sequencing (WES) in a trio scheme (the proband and both parents).

## 3. Genetic Studies

DNA from both probands and their relatives was extracted from blood using a standard protocol. Both patients were subjected to exome sequencing (ES). For proband 1, a singleton analysis was performed using SureSelectXT Human All Exon v7 (Agilent Technologies, Cedar Creek, TX, USA), while for proband 2, a trio analysis was performed using Twist Human Core Exome 2.0 + Comp Spike-in + Twist mtDNA Panel (Twist Bioscience, South San Francisco, CA, USA). The enriched libraries were pair-end sequenced (2 × 100 bp) on NovaSeq 6000 (Illumina, San Diego, CA, USA). A bioinformatics analysis of the raw WES data and variant prioritization were performed as previously described [8]. Identified variants were further annotated with functional information, frequency in population (including the gnomAD database http://gnomad.broadinstitute.org/, accessed on 10 August 2023, and an in-house database of >8500 Polish exomes), and known association with clinical phenotypes based on both ClinVar (https://www.ncbi.nlm.nih.gov/clinvar/, accessed on 10 August 2023) and HGMD (http://www.hgmd.cf.ac.uk/, accessed on 10 August 2023) databases. In silico pathogenicity prediction was performed based on VarSome-provided pathogenicity and conservation scores [9] and the American College of Medical Genetics and Genomics (ACMG) guidelines [10]. For proband 1, variants considered as potentially disease-causative were further subjected to a family study and were analysed in the proband, his heathy parents and brother by amplicon deep sequencing (ADS) using a Nextera XT Kit (Illumina) and sequenced) as described above. For proband 2, the inheritance of plausible disease-causing variants was determined directly based on the trio ES analysis.

In both probands, missense variants in the *ZMYND11* gene were identified and prioritized as causative: NM_001370100.5:c.1262G > A, NP_001357029.1:p.Ser421Asn, rs869320713 located in exon 13 (position 35 of 273) in proband 1, and a novel NM_001370100.5:c.1253T > G, NP_001357029.1:p.Val418Gly located in exon 13 (position 26 of 273) in proband 2. For both p.Ser421Asn and p.Val418Gly, *ZMYND11* variants were absent in the probands’ parents and were considered as likely de novo events.

The p.(Ser421Asn) variant has 0 frequency in the gnomAD v3.1.2 database (accessed on 10 August 2023) and is classified as a variant of uncertain significance (VUS) according to the ACMG guidelines (score: 3 points, PP5 Moderate, PM1 Supporting, PM2 Supporting and BP4 Supporting). However, when the de novo status is taken into account, the variant is classified as likely pathogenic (score: 7 points, PP5 Moderate, PM1 Supporting, PM2 Supporting, BP4 Supporting and PS2 Strong). The results of the in silico pathogenicity prediction are as follows: a CADD score of 24.1, six meta-score predictors indicate p.Ser421Asn as “benign”, six individual predictions as “pathogenic”, six as VUS and 13 as “benign”. The p.Ser421Asn variant is located in a highly constrained coding region (CCRS = 92.33) and the c.1262G is highly conserved (phyloP100: 9.646). Moreover, the p.(Ser421Asn) variant is registered in the ClinVar database (version: 31-July-2023) and classified as “likely pathogenic” (accession no. VCV000225254.2); in the HGMD database, it has a confidence of “high”, and has been reported as disease-causing in patients with neurodevelopmental disorders [11,12,13].

The p.Val418Gly variant has 0 frequency in the gnomAD v3.1.2 database (accessed on 10 August 2023) and is classified as VUS according to the ACMG guidelines (score: 3 points, PM1 Moderate and PM2 Supporting). However, when the de novo status is taken into account, the variant is classified as likely pathogenic (score of 7 points, PM1 Moderate, PM2 Supporting and PS2 Strong). The results of the in silico pathogenicity prediction are as follows: a CADD score of 28.5, three meta-score predictors indicate p.(Ser421Asn) as “pathogenic”, one as VUS, five as “benign”, three individual predictions as “pathogenic”, eight as VUS, and ten as “benign”. The p. Val418Gly variant is located in a highly constrained coding region (CCRS = 92.33) and the c.1253T is highly conserved (phyloP100: 7.736).

## 4. Discussion

Histone modifications (acetylation, methylation and others) affect chromatin structure, provide binding platforms for diverse transcription factors and, therefore, play important roles in many cellular events, including gene expression, DNA repair and cell cycle control [14]. Misregulation of histone modifications is associated with developmental defects [15]. The Zinc finger MYND domain-containing protein 11 is an epigenetic reader protein that binds to a specific methylated histone (H3.3K36me3) to regulate pre-mRNA [16]. Disorders of transcriptional regulations associated with histone modification and chromatin remodelling, such as the Coffin–Siris syndrome (disruption of histone phosphorylation), the Rubinstein–Taybi syndrome (disruption of histone acetylation), the Sotos syndrome (disruption of histone methylation) and the Cornelia de Lange syndrome, display similar clinical features to the *ZMYND11* disease, such as developmental delay/intellectual disability, facial dysmorphisms and growth problems [17]. In a group of 57 patients with a clinical suspicion of Cornelia de Lange syndrome, one *ZMYND11* frameshift variant was identified [18]. The *ZMYND11* gene is expressed in many human tissues, including the brain. Pathogenic variants in the *ZMYND11* gene can lead to a broad spectrum of signs and symptoms. There is considerable variability in the degree of psychomotor and speech developmental delay and intellectual disability in patients with this syndrome [6,7]. The patients presented in this paper were referred for a genetic examination due to severe developmental delay, hypotonia and dysmorphic features. They were identified through a whole-exome sequencing test.

Pathogenic variants in the *ZMYND11* gene causing a specific dysmorphic syndrome were first described in 2014 [2,5]. Further cases were reported by Moskowitz et al. in 2016 [11] and by Popp et al. in 2017 [19]. Yates at al. [6] included in their paper 27 patients with pathogenic variants in the *ZMYND11* gene. All of them had intellectual disabilities, including 4 with severe intellectual disability (21/21, except for 6 in whom no information was provided), 38% had epilepsy and 48% had hypotonia. Growth abnormalities like a short stature (proband 1) or microcephaly (proband 2) are additional findings for this disorder.

It has been described that patients with truncating variants in the *ZMYND* gene have a similar phenotype to the 10p15.3 microdeletion syndrome, but in some missense variants, the phenotype is more severe (for example, the cases described by Moskowitz et al. and Cobben et al. [5,6,11]).

Proband 1 and a 24-year-old female described by Moskowitz et al. have the same variant in the *ZMYND11* gene. Both patients had global developmental delay, especially affected speech, hypotonia and delayed myelination in the MRI. However, unlike the case of the patient described by Moskowitz et al. [11], in the patient described in this paper, there was no evidence for us to diagnose epilepsy, although our patient had been receiving an antiepileptic drug for several years. The clinical characteristics of the presented patients and the literature references are shown in Table 1.

Feeding problem is another issue affecting a significant proportion of people with this *ZMYND11* variant. This finding may be partially associated with global hypotonia. In proband 2 presented here, the following symptoms appeared with regard to this matter: choking during and after a meal, food retraction, spitting up, and increased gag reflex after the intake of solid foods. A similar clinical picture has already been described in the literature. Yates et al. [6] reported that problems with feeding, including excess vomiting after feeds and/or bottle feeding requiring more than one hour, were present in 59% of patients described in their work. However, the manifestation was usually mild, with only three patients requiring supplementary feeding by a nasogastric tube [6]. Interestingly, two patients described in the literature presented with tooth enamel hypoplasia. A problem that we observed in our patient was delayed eruption of the teeth, which is not extensively discussed in the literature. When compared to other works, it is noticed that no patient presented with hypoglycaemia.

HH is characterised by hypoglycaemia with an inadequate suppression of insulin, detectable C-peptide, a suppressed or low level of ketones (3-β-hydroxybutyrate < 2 mmol/L) and an absence of metabolic acidosis [20]. The time of onset of hypoglycaemia, its correlation with the type of meals, and the presence of dysmorphic features are essential because HH is a heterogenous disorder that is divided into a transient form related to perinatal stress or following gastrointestinal procedures, and a congenital, monogenic HH form related to variants in fifteen genes implicated in pancreatic development and function (due to a genetic defect, insulin release is independent from the glucose level) and associated with other syndromes (such as Beckwith–Wiedemann syndrome) [21,22]. Although the loss of function of the *ZMYND11* gene is a recognised cause of hypotonia and feeding problems, it has not previously been noted as a risk factor for HH. We studied the exome sequencing data of the presented patients and did not find any pathogenic or likely pathogenic variants that could cause HH. However, these are not rare cases because a large number of HH cases remain unexplained. In approximately 50% of patients with the persistent form of HH, a known underlying genetic basis may explain the molecular mechanism of dysregulated insulin secretion [20]. In proband 2, the rather late onset of hyperinsulinaemic hypoglycaemia (in the 15th month of life) and the unnatural way of feeding via a gastrostomy tube are noteworthy. Postprandial hypoglycaemia is a common complication of Nissen fundoplication in children as well. The mechanism responsible for hypoglycaemia involves an exaggerated secretion of the glucagon-like peptide 1 (GLP-1), which may contribute to excessive insulin secretion and subsequent hypoglycaemia [23]. The fact that the proband received only liquid meals after the surgery might have further exacerbated the problem because it is known that a liquid meal results in a significantly higher GLP-1 release than a solid meal of an identical composition [24]. However, hypoglycaemia (30 mg/dL) was diagnosed 2 days before the gastrostomy insertion, but diagnostics were not continued at that time. The question remains whether, in the presented patients, pathogenic variants in the hitherto unknown genes responsible for HH cause the disorder, or whether it is the result of environmental or epigenetic factors. A successful attempt to return to the natural way of feeding could provide the answer, but so far, this is not possible. As a result of inserting the gastrostomy tube, the frequency of recurrent pneumonia with respiratory failure was reduced. The non-physiological way of feeding could, however, contribute to the appearance or aggravation of HH. This was the reason for the introduction of diazoxide, which, in turn, might have exacerbated the hypotonia [25]. Diazoxide is the first-line agent for the treatment of congenital HH. If a patient is responsive to diazoxide, then they could be treated with this medicine for a long time. Common side effects of diazoxide are mild-to-severe hypertrichosis (as in our patient), fluid retention, pulmonary hypertension, neutropenia and thrombocytopenia. Diazoxide inhibits insulin release, but can also cause smooth muscle relaxation by opening K ATP channels (ATP-sensitive potassium channel), which leads to cell hyperpolarisation and, thus, the blockage of calcium channels [26]. However, it is worth noting that the opening of KATP channels has a less significant impact on the regulation of electrical activity of muscle and nerve cells, which results from the contribution of additional channels to the resting membrane potential and membrane resistance in those tissues [27].

One of the most severe issues of proband 2 are the pathologies in her brain, as revealed by the MRI examination performed at the age of 14 months. The brain MRI showed an absence of normal myelination according to the age of the child, which suggested delayed myelination of the white matter and indicated a stage of myelination typical for the age of 8 months. There was also dilatation of the lateral ventricles, the third ventricle and the cerebral sulci as a sign of cerebral cortico-subcortical atrophy.

The clinical picture suggests two main causes for these disorders: hypoglycaemia and the presence of the *ZMYND11* mutation. In the case of both diagnoses, CNS abnormalities of various morphologies are very common. In the case of hypoglycaemia, disturbances in the structure of the cerebral cortex are described in the literature, including advanced destructive changes in the white matter and dilatation of the ventricular system [28,29], which are consistent with the MRI image of our patient. In addition, intraventricular haemorrhage (IVH) reported by two centres [30,31], both occurring within the first 10 days of life, seems to be associated with HH, but not in the proband presented in this paper, as we did not observe any low signals of hemosiderin depositions in the SWI sequence. We believe that the vanishing syndrome (VWM) is worth mentioning in the context of the described proband, although the typical MR appearance of white matter alterations in the VHM syndrome is completely different. The characteristic MRI findings of VWM include diffuse, symmetric cerebral white matter involvement with T2 and FLAIR high-signal-intensity areas, which extend from the periventricular regions to the subcortical arcuate fibres. Over time, the white matter vanishes and is replaced by characteristic areas presenting almost CSF signal intensity, which means low signal changes on FLAIR images [32]. Cerebellar atrophy is also present in VWM, being the only similar feature regarding the white matter MR appearance in patients with VWM and in our patient.

Moving onto neurological disorders in cases with ZMYND11 pathological variants, their range is immensely wide and includes generalised developmental delay, seizures, autism and behavioural abnormalities [6,7,11]. Among them, the presented patient 2 suffers from generalised developmental delay and seizures, which were diagnosed at 2 months of age. We believe that the *ZMYND11* mutation also seems to have a significant impact on dysmyelination changes in the presented probands, which is confirmed by scientific reports. Cortico-subcortical atrophy of the cerebral cortex [1], cerebral atrophy, delayed myelination and compression of myelin [5,7,11] are MRI findings that have been reported in the papers published hitherto. These vastly coincide with the changes exhibited by our probands.

## 5. Conclusions

In summary, we reported two new probands with *ZMYND11* variants and described the clinical and radiological features that are worth mentioning due to their significant relevance for current knowledge, as well as their contribution to a better understanding of the symptoms and genotype–phenotype correlation of this emerging disease. Based on the cases presented, we wanted to point out that the phenotype of patients with a mutation in the ZMYND11 gene may play a significant role not only in the development of severe hypotonia but also in feeding problems. It is also possible that the HH seen in patient 2 is coincidental and has no connection with the pathogenesis associated with the ZMYND11 gene.

## Figures and Tables

**Figure 1 genes-15-00256-f001:**
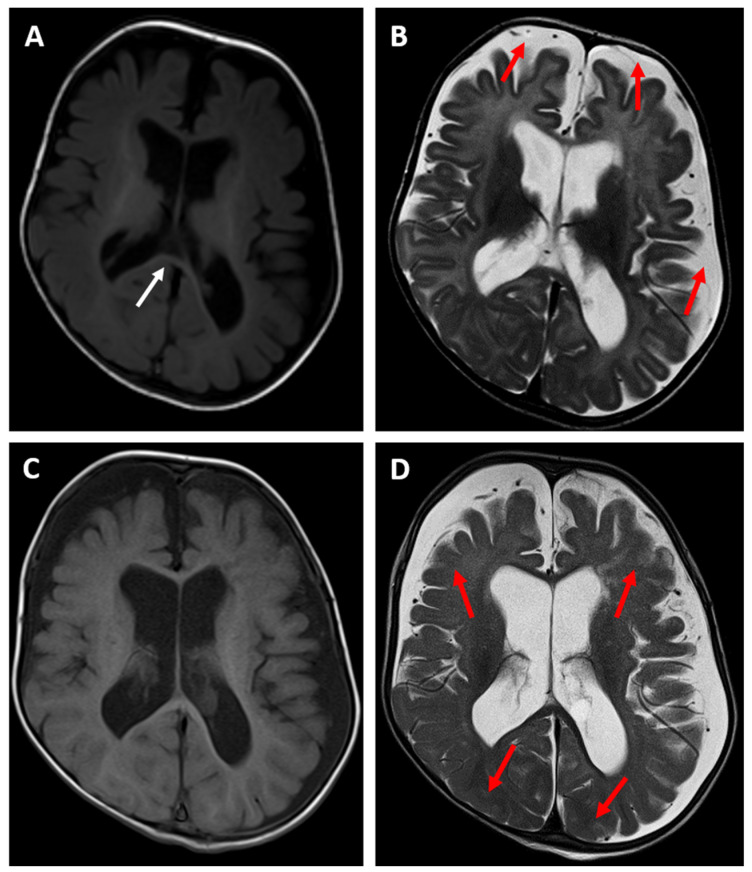
Brain MR examinations of patient 1: axial T1-weighted images (**A**,**C**) and axial T2-weighted images (**B**,**D**) performed at the age of 6 months (upper row) and at the age of 14 months (bottom row). The images show a feature of cortico-subcortical atrophy with an adjacent enlargement of the subarachnoid space in the fronto-temporal areas ((**A**,**B**)—red arrows). There is also delayed myelination of the white matter, indicating a typical pattern for the age of 4 months ((**A**)—white arrow pointing to a high signal intensity of the splenium of the corpus callosum on the T1-weigthed image). The follow-up MRI examination (**C**,**D**) revealed progression of the myelination process; the cortex/white matter differentiation is blurred, especially on the T2-weighted image ((**D**)—red arrows), instead of being clearly visible at this age, which can indicate a demyelination disorder.

**Figure 2 genes-15-00256-f002:**
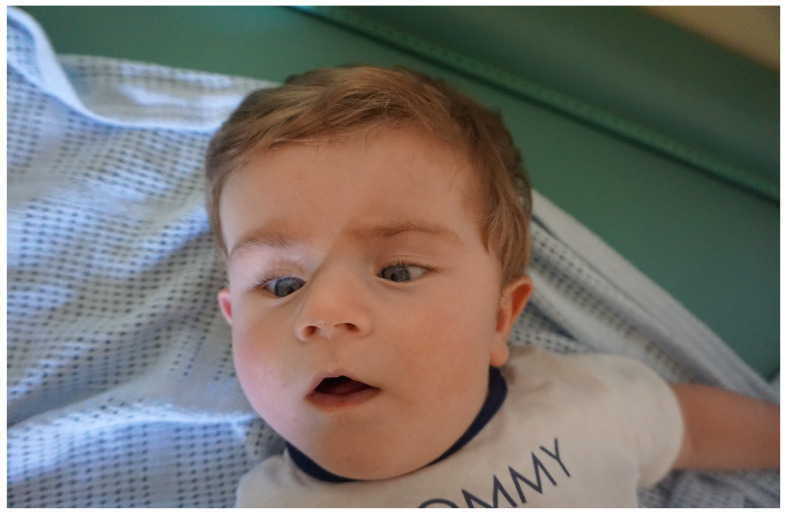
Facial phenotype of patient 1 at 6 months of age.

**Figure 3 genes-15-00256-f003:**
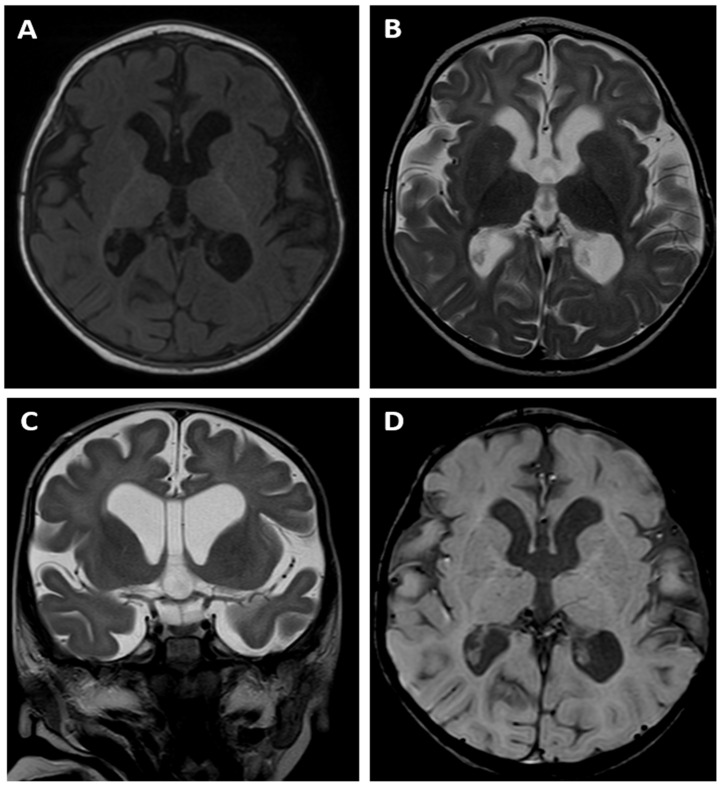
Brain MR examination of patient 2: axial T1-weighted image (**A**), axial (**B**) and coronal (**C**) T2-weighted images, as well as SWI image (**D**). The T1- and T2-weighted images (**A**–**C**) suggest delayed myelination of the cerebral white matter, indicating a stage of myelination typical for the age of 8 months. There is also dilatation of the ventricular system and cerebral sulci as a sign of cerebral cortico-subcortical atrophy. The MRI also showed a cavum septum pellucidum (**C**). The SWI sequence (**D**) did not reveal any low signals of hemosiderin depositions.

**Figure 4 genes-15-00256-f004:**
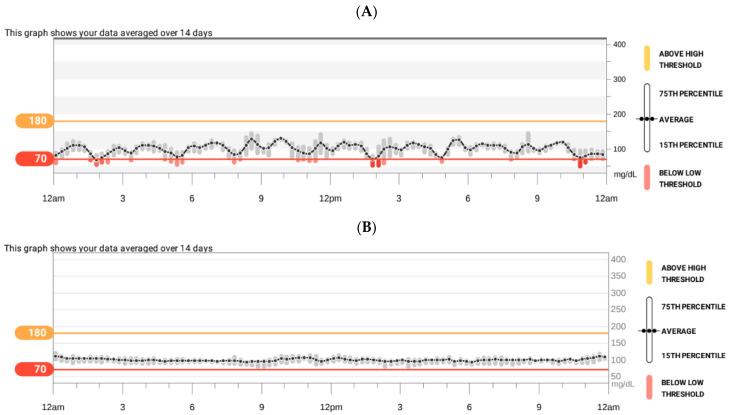
Results of continuous glucose monitoring in patient 2: at diagnosis of hyperinsulinaemic hypoglycaemia (**A**) and during treatment with diazoxide (**B**).

**Figure 5 genes-15-00256-f005:**
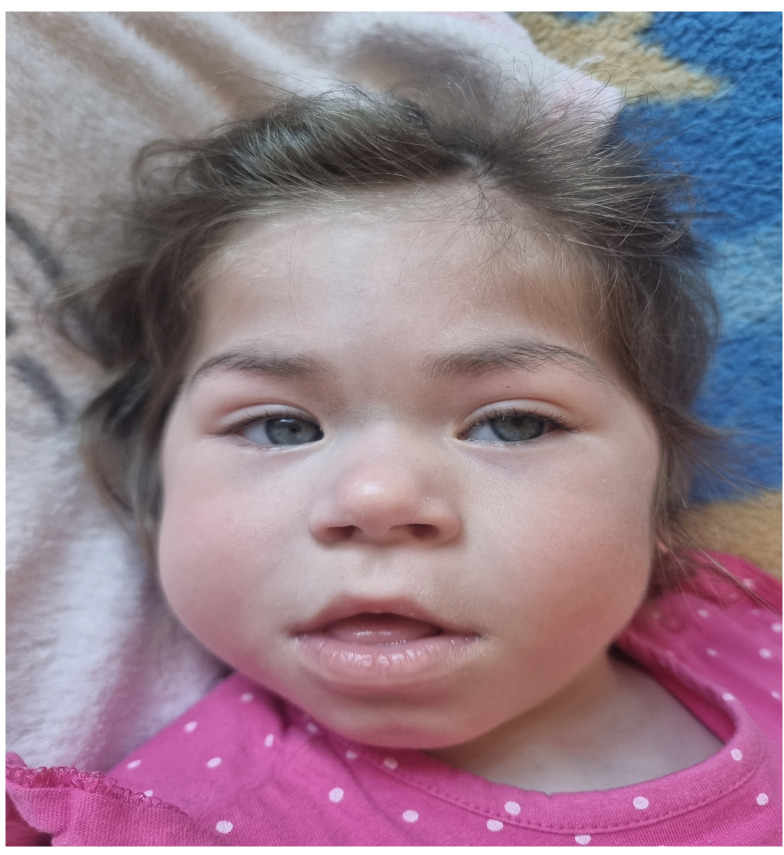
Facial phenotype of patient 2 with a ZMYND11 variant at 22 months of age.

**Table 1 genes-15-00256-t001:** Clinical characteristics of presented patients and referred to the literature.

	Proband 1	Proband 2	Moskovitz et al. [11]
age reported/gender	5.6 yr/male	2,5 yr/female	24 yr/female
birth weight (SDS) length (SDS)	4400 g (+2.39) 57 cm (+4.03)	4200 g (+1.94) 58 cm (+4.77)	3740 g 52 cm
HBD	37	38	born full term
birth OFC (cm)	37 (2.14)	34 (0)	-
OFC (age reported)	53.8 cm (75–90 c)	45.5 c < 3 centile (1.8 year)	microcephaly 51.5 cm
height/weight	short stature	normal height	no information
feeding problems	mild	severe nasogastric tube feeding; at 12 months, PEG was placed	severe eosinophilic esophagitis
development: gross motor	sitting at 24 mo. and walking at 4–5 yr.	sitting, walking not yet achieved	sitting at 18 mo. and walking at 4–5 yr.
development: speech/communicate	nonverbal, but makes sounds.	nonverbal	nonverbal, but makes sounds/communicates using a few signs
hearing impairment	no	mild	mild
hypotonia	yes	yes	yes
epilepsy	no	yes (diagnosed at 2 mo.)	yes (diagnosed at 9 mo.)
brain MRI (age at which the test was performed)	cerebral atrophy and delayed myelination (6 and 14 mo.)	cerebral atrophy and delayed myelination (14 mo)	cerebral atrophy and delayed myelination (8 years)
behavioural difficulties	hyperactivity with sleeping problems		hyperactivity with happy disposition
dysmorphic	deep-set eyes, long and thick eyebrows, strabismus, consistently open mouth, low-set ears with prominent auricles	prominent eyebrows, strabismus, consistently open mouth, small teeth, low hairline	deep-set eyes, prominent jaw, wide mouth, broad nasal root and bulbous nasal tip,
frequent infections	recurrent respiratory infections, otitis media, and encephalitis in the course of Streptococcus pneumoniae infection	recurrent lower respiratory infections and chronic otitis media	chronic otitis media, sinusitis, and recurrent lower respiratory infections
allergy	allergy to cow proteins was suspected.	allergy to cow proteins was suspected	eosinophilic esophagitis, asthma, atopic dermatitis, and drug and food allergies
variant in ZMYND11 gene	c.1262G > A	c.1253T > G	c.1262G > A
variant mechanism	missense	missense	missense
predicted effect on protein	p.Ser421Asn	p.Val418Gly	p.Ser421Asn

## Data Availability

The data presented in this study are available on request from the corresponding author. The data are not publicly available due to ethical reasons.

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
