# Peer review of "Further Delineation of Clinical Phenotype of ZMYND11 Variants in Patients with Neurodevelopmental Dysmorphic Syndrome"

_genes, 2024, doi:10.3390/genes15020256_

Round 1

Reviewer 1 Report

Comments and Suggestions for Authors

Overall comments:

This paper describes the disease phenotypes of two subjects with ZMYND11 missense variants.  The data are sufficient to support the conclusion that the syndromes in both subjects are due to the ZMYND11 mutations, although it would have been helpful had data been included on other unique sequence variants were found in the probands.  Numerous ZMYND11 mutation cases have been published previously and the phenotypes of the probands described in this study overall do not differ significantly from those of the other described cases.  An exception is the presence of hyperinsulinemic hypoglycaemia (HH) in one of the 2 probands in this study.  The authors suggest this may be part of the ZMYND11 mutation syndrome.  However, there are no data to support this hypothesis, and it seems unlikely since HH has not been seen in other subjects with a variety of ZMYND11 mutations. This, overall the paper makes a minor contribution to our knowledge of the phenotypic spectrum of the ZMYND11 mutation syndrome.

Specific Comments:

Lines 27-28: The data do not justify saying “Although the reason for the occurrence of HH in our proband is not clear, it is likely that it broadens the spectrum of symptoms characteristic of ZMYND11 syndrome.”  Since this sign has not been reported in other cases, it could be coincidental. 

Line 38:  The number 11 appears to be out of place here.  It should be deleted.

Line 42: Replace “has been” with “have been”.

Line 43: Replace “cause” with “causes”.

Line 44: What does MRD30 stand for?

Lines 46-47: Change “… craniofacial dysmorphism results in a phenotype resembling 10p15.3 microdeletion syndrome …” to “craniofacial dysmorphism. This phenotype resembles 10p15.3 microdeletion syndrome …”

Line 48: insert “a” before “splice variant”.

Lines 58-64: Delete all but the first sentence of this paragraph.  It is not appropriate to summarize the results in the introduction.

Line 67: The word “year” is missing from the age.

Lines 74-75: “recurrent” and “repeatedly” are redundant.  Delete one of these.

Line 77: Make it clear whether you mean the subject was first able to sit up and walk at the indicated ages.

Lines 83-84: Were the meningitis and encephalitis due to Streptococcus pneumoniae infection treated? How?

Line 87: Replace “the history” with “a history”.

Line 101: Change “indicated” to “indicate”.

Lines 109-114: Give normal reference ranges for glucose and insulin levels.

Line 115: For age, insert “years”.

Line 121: Change “are note” to “were noted”.

Figure 1: Label the important features.

Line 141: Change “in” to “at”. Delete ZMYND11.

Line 208: Provide reference range for blood glucose.

Lines 218-221: Need to provide reference ranges for all parameters.

Line 245: What does HH stand for?

Line 281: “variant” should be plural – “variants”.

Line 282: Change to “potentially disease causative”.

Lines 287 - : Need to provide lists of the variants that were present in the probands but not in the reference population.  Typically, there are numerous candidate variants.  It is not clear how other potential causal variants were excluded. It is not clear whether the variants were heterozygous.  This needs to be stated clearly.  How many other unique heterozygous variants in other genes were present in the probands? 

Discussion: Grammar in this section needs editing.

Lines 337-340: References need to be cited to support this statement: “Pathogenic variants in the ZMYND11 gene can lead to a broad spectrum of signs and symptoms. There is considerable variability in the degree of psychomotor and speech developmental delay 339 and intellectual disability in patients with this syndrome.”

Lines 351-352: This statement is not informative unless a description of 10p15.3 microdeletion syndrome is provided: “It was noticed that patients with truncating variants in ZMYND gene have a similar phenotype to 10p15.3 microdeletion syndrome …”

Line 355: Is “Presented first patient” proband 1?  This should be stated.

Lines 355-361:  It does not appear that the description of proband 1 adds anything new to the description of the disease phenotype of the subject with the same variant that was described in the Moskowitz et al. paper.

Lines 375-427: Most of this discussion of HH is speculative and does not anything.  This paragraph should be shortened to indicate that no variants in genes associated with HH were identified in the proband, so the cause of HH remains unknown.  Since it has not been reported in any other cases, it is likely coincidental.

Lines 428-433:  This is just a restatement of the results. 

Lines 434-451: This discussion does not take into account that other cases of ZMYND11-associated disease are not accompanied by HH, again suggesting that this could just be a coincidental finding.

Lines 453-459: This discussion does not appear to relevant and could be deleted.

Comments on the Quality of English Language

There are numerous grammatical errors that need to be corrected.  I have pointed out many of these, but further editing is required to correct additional errors.

Author Response

Dear Reviewer,

Reviewer 2 Report

Comments and Suggestions for Authors

In this paper, the authors present very detailed case reports of two previously undescribed patients with alterations in the ZMYND11 gene. Variants of this gene are well known to cause “autosomal dominant intellectual developmental disorder-30 with speech delay and behavioral abnormalities (MRD30)” (OMIM 616083; ClinGen https://search.clinicalgenome.org/kb/gene-dosage/HGNC:16966), and the characteristics of these patients are consistent with this molecular diagnosis. The strength of this paper is in presenting the detailed and meticulous clinical description and phenotyping of the cases.

Overall, the genomic analyses presented are fine. The methods applied are relatively standard. Trio WES is performed for one case, and singleton WES (referred to as “mono analysis” – which is to my knowledge not a standard term and should probably be avoided) in the other, with secondary confirmation by PCR and Sanger sequencing in the parents and proband. Evidence for (likely) pathogenicity of these variants seems adequate.

A central point of the paper that attempts to distinguish it from being a simple set of case reports is that persistent hyperinsulinemic hypoglycaemia (HH), seen in one of the patients, may be a previously unrecognized feature of impaired ZMYND11 function. This hypothesis is, however, not supported by the data presented. What plausible link is there between the ZMYND11 protein and its known functions and this clinical condition? The fact that it is seen in this patient, in the absence of any other detectable genetic etiology, is insufficient. Indeed, it is even stated (lines 391-394) that only 50% of patients with HH have a known (or suspected?) genetic etiology – meaning that such a cause is unknown in the remaining 50%. While this figure is based on a 2017 article, and may now be different, nevertheless it is most likely that the HH seen in this patient is coincidental, has nothing to do with ZMYND11, and has a large chance of not being due to any detectable single gene cause. It should not be implied that HH might be a previously unrecognized expansion of the ZMYND11/MRD30 syndrome, and making such an assertion in the absence of any firm, supportable, and replicated evidence should not be encouraged.

Finally, only two cases are reported here. Given the comparatively larger number of individuals reported in the literature to date, the addition of these two seems unremarkable and does not contribute much to furthering knowledge of this condition.

Minor suggestions:

Methods references should be referred to bibliographically, not as PMIDs.

There are some formatting problems with Figure 6, but this figure is probably not necessary.

Comments on the Quality of English Language

There are some minor grammatical issues, mainly with pluralization, but nothing of any great concern.

Author Response

Dear Reviewer,

Round 2

Reviewer 1 Report

Comments and Suggestions for Authors

The critiques have been adequately addressed.  

Comments on the Quality of English Language

Minor grammatical editing required.

Reviewer 2 Report

Comments and Suggestions for Authors

I thank the authors for their careful considerations of my previous points. As a report of two cases of ZMYND11/MRD30 syndrome, the paper is fine and would be of interest to others researching or caring for individuals with this disorder.